# Impaired Affordance Perception as the Basis of Tool Use Deficiency in Alzheimer’s Disease

**DOI:** 10.3390/healthcare10050839

**Published:** 2022-05-02

**Authors:** Nam-Gyoon Kim, Judith A. Effken, Ho-Won Lee

**Affiliations:** 1Department of Psychology, Keimyung University, Daegu 42601, Korea; 2College of Nursing, University of Arizona, Tucson, AZ 85724, USA; jaeffken@comcast.net; 3Department of Neurology, School of Medicine, Kyungpook National University, Daegu 41404, Korea; neuromd@knu.ac.kr

**Keywords:** Alzheimer’s disease, affordance, apraxia of tool use, alternative tool use, Go/No-Go task, preclinical stage

## Abstract

The present study investigated whether defective affordance perception capacity underpins tool use deficits in patients with Alzheimer’s disease (AD). An affordance, a concept James Gibson introduced, scales environmental objects to an animal’s action capabilities, thus offering opportunities for action. Each man-made artifact carries both a primary affordance (its designed function) and secondary affordances. In Experiment 1, participants identified secondary affordances of objects as a measure of their ability to identify alternative uses of familiar tools. A single response Go/No-Go task was administered to 4 groups: AD, mild cognitive impairment (MCI), Parkinson’s disease (PD), and elderly controls (EC). Groups were matched for age and years of education. The AD group performed poorest, followed by MCI, and PD and EC. EC and PD groups’ results failed to reach statistical significance, and the AD group performed at chance. In Experiment 2, participants judged the physical properties of the same objects used in Experiment 1. Even AD patients performed reliably, ruling out a visual processing deficit as the basis for their poor performance in Experiment 1. Results suggest that degraded affordance detection capacity can differentiate AD from normal aging and other neurodegenerative disorders and could be an affordable marker for AD, even in the early stages of AD.

## 1. Introduction

Alzheimer’s disease (AD) is a progressive and irreversible degenerative disease that destroys memory and other intellectual functions such as reasoning, planning, and language. As the disease advances, patients with AD gradually lose their capacity to perform essential activities of daily living, eventually necessitating their reliance on caregivers. With increased dependence on caregivers for their daily activities, patients become less autonomous, and their quality of life deteriorates. Caregiving is expensive, particularly for AD patients. The increasing demand for caregiving will inflict a heavy financial strain on the public health system, but its impact on individual patients, their families, and caregivers will be even more severe because of the physical, emotional, and financial stresses incurred [1].

According to the World Health Organization (WHO), over 55 million people worldwide (8.1% of women and 5.4% of men over 65) suffer from some form of dementia (2 September 2021). People live longer now than ever before. Given the extended life expectancy, the prevalence of dementia is expected to triple by 2050. Among dementias, AD is the most prevalent, accounting for 60–70% of the cases. With the majority of those individuals living in low- and middle-income countries, the social and economic burden of the disease is likely to pose significant challenges, particularly for these countries [2].

Despite enormous resources and extensive efforts devoted over the last few decades, the etiology of AD remains elusive and the cure for the disease has yet to be discovered. Recent breakthroughs in biomarker research characterize AD as a continuous process that begins 15 to 20 years before cognitive changes appear [3,4,5]. This long period (termed *preclinical*) is now used to classify individuals with normal cognition but with biomarker evidence of AD pathology [6,7,8]. Based on the degree of cognitive impairment identified, AD is now subdivided into three stages: preclinical AD, mild cognitive impairment (MCI) due to AD, and dementia due to AD [6,7,8].

Given the absence of effective AD treatments, the current consensus is that, if disease-modifying therapy is to be effective, the drug or treatment intervention must occur during AD’s preclinical stage when pathophysiological processes are present but have not yet produced identifiable cognitive impairments. Coincident with the current shift in AD research from treatment and cure to prevention and risk reduction, research objectives too are shifting toward identifying novel techniques to screen more carefully those individuals who are cognitively normal but display underlying evidence of AD pathology so that available therapeutic interventions can be begun promptly to delay, or even prevent, debilitating neural deterioration [9,10].

Diagnosing AD has traditionally relied on clinical and neuropsychological evaluations plus brain imaging and laboratory tests to exclude other causes of dementia. However, in the preclinical phase of AD, the underlying pathophysiological process remains silent, not yet having triggered cognitive alterations. Although the biomarkers developed to date are capable of identifying preclinical AD [11,12], these biomarkers can only be obtained using lumbar puncture and PET scans. Both procedures are expensive, invasive and not widely accessible [13,14] (also see [15] for theoretical limitations of current biomarkers). For these reasons, these biomarker tests are not widely used in routine clinical practice. Thus, there is an urgent need to develop less invasive, less expensive, and more easily accessible biomarkers that can detect AD in the preclinical stage. 

Recent research has demonstrated that, during the supposedly asymptomatic phase of AD progression, the pathophysiological process of AD appears to cause subtle degradation in the individual’s cognitive performance [8,16,17,18]. However, the extent of the degradation is insufficient to meet current criteria for MCI. Neuropsychological tests have been shown to detect subtle cognitive alterations. If the pathophysiological process of AD leaves detectable traces, even in the early stage of AD progression prior to converting to MCI, neuropsychological testing could play a more significant role as a diagnostic tool for AD in all phases of the disease. Neuropsychological tests have the benefits of being non-invasive, inexpensive, easily available, and reliable diagnostic tools. Given their potential capacity to probe into the preclinical phase of AD progression, neuropsychological testing will continue to be an important staple of diagnostic work-up for AD. In fact, neuropsychological evaluation is indispensable for assessing disease progression, evaluating treatment effects, and validating biomarkers [18,19,20,21,22,23]. 

Identifying and testing biomarkers for AD is evolving rapidly, but neuropsychological evaluation has continued to use a 60-year-old paradigm that primarily focuses on episodic memory impairment [24]. The results of current AD biomarker research suggest that cognitive assessment should incorporate diverse measures that are sensitive enough to detect and track subtle changes, even in the asymptomatic stages of the disease [25,26]. In the present study, we explored whether praxis disturbance, a largely neglected aspect of AD, can be utilized to augment the diagnostic power of existing neuropsychological tests so they are sensitive enough to detect subtle cognitive changes occurring in the asymptomatic stages of AD and, at the same time, specific enough to differentiate AD from other neurodegenerative disorders. 

Praxis, meaning “doing, acting” in Greek, refers to the ability to perform skilled or learned movements. When this capacity is disturbed (a disorder known as apraxia), the affected individual is unable to perform skilled movements in response to a verbal command despite having adequate sensory and motor abilities and comprehension of the task. Liepmann, who introduced the term in the early 1900s, described apraxia as a disorder of motor control, based on his observation of a patient who suffered a left hemispheric stroke [27,28]. Liepmann conceived praxis (i.e., skilled movements) as a two-stage sequence consisting of conception and production, which has evolved into a conceptual-production systems model [29]. During the conception stage, the image (i.e., concept) of the intended action is constructed based on a movement formula (“visual engrams” of the action) retrieved from the parietal lobe of the dominant (usually left) hemisphere. During the production stage, the action image is conveyed via a stream from the parietal areas to the frontal areas where it is converted into the motor commands (i.e., movement-specific signals) necessary to recruit muscles to carry out the intended action [27,30,31,32]. 

Liepmann classified apraxia into subtypes, two of which are the most representative and are demonstrated by selective impairment of either the conceptual or the production system. As noted above, Liepmann hypothesized that movement formulae (i.e., abstract representations specifying the spatial and temporal sequence of movements composing an action) are stored in the left parietal lobe. If the left parietal lobe is damaged, the movement formula for the intended action may also be damaged or its activation disrupted. Thus, an individual, now unable to conceive the action intended, may perform the wrong movement (e.g., combing hair with a toothbrush) or perform a sequence of movements in the wrong order (e.g., pouring the water before opening the coffee maker) [33,34]. This “wrong movement” condition is referred to as ideational apraxia (IA). 

Once the movement formula is retrieved, it is transferred to the motor cortex via the posterior-anterior stream where it is converted into the motor commands that will guide movement implementation. However, if this stream is disrupted, the idea of an action developed by the conceptual system is dissociated from its execution. When asked to carry out an action or imitate one, the individual, although fully aware of the task, executes the action in an odd and clumsy fashion. This condition is referred to as ideomotor apraxia (IMA).

One significant aspect of apraxia is that it affects a person’s ability to use commonly available tools or adapt objects in the surrounding environment to serve as tools to solve a given problem. Indeed, tool use is considered as one of the defining features of humans. Thus, whether a task is required for self-care and self-maintenance or for living independently, impaired ability to use tools will limit an individual’s functional capacity. It is commonly understood that a tool is a man-made implement designed to perform a particular function. As a physical object, a tool must be manipulated in a specific way to maximize its function. For proper tool use, its user must know both what the tool is designed to do (function knowledge) and how to manipulate the tool (manipulation knowledge). 

Until recently, semantic knowledge of object function was thought to be instrumental in tool use. Semantic memory refers to long-term memory containing general world knowledge accumulated throughout one’s life. Thus, when an individual intends to use a tool, he retrieves from semantic memory the stored representation of tool function, i.e., the purpose, its action recipient, and the prototypical movements associated with the tool (e.g., a screwdriver is used to tighten or loosen a screw and is typically kept in the toolbox; To fulfill the tool’s purpose, grasp the handle with the dominant hand and turn the screwdriver clockwise to tighten the screw, etc.) [35,36,37]. 

However, in everyday interactions with our surroundings, we are confronted with situations necessitating our use of familiar tools for unconventional purposes (e.g., using a coin to drive a screw) or the use of unfamiliar tools for which there are no associated semantic memories [38]. Semantic knowledge (when understood as a repository of information about canonical movements associated with familiar tools) is of little value in these situations. The role of semantic knowledge related to tool use has been investigated in patients with semantic dementia (SD) [39,40,41]. SD is a neurodegenerative disorder, primarily involving comprehension of words and related semantic processing, but progressing ultimately to lost comprehension of objects. In the investigations cited above, SD patients performed poorly in naming familiar objects and identifying those objects’ functions. However, when asked to manipulate the same objects used in the naming task, they performed with ease and without hesitation. However, Riddoch et al. [42] reported the opposite pattern involving an apraxic patient with intact object identification and object naming but with severe deficits in object use (see also [43,44,45] for similar findings). 

These contrasting performance patterns suggest that conceptual knowledge and object use are likely dissociable processes subserved by separate neurological systems. Indeed, these findings corroborate the rationale underpinning the dual route model of action production proposed by Roy and Square [29] (see also [34,46,47]). 

The dual route model postulates two complementary and interacting routes between visual input and motor output. For both routes, the process begins with sensory analysis of the tool and terminates with execution of the action. Sensory analysis involves extracting the structural properties of the tool from sensory information (signals from sensory receptors activated by environmental stimuli). In the semantic (or indirect) route, extracted information is then processed using semantic knowledge stored in long-term memory in which the function of the tool (functional knowledge) and the prototypical manner of handling the tool (manipulation knowledge) are retrieved and then combined to select the relevant movement parameters to be implemented by the action system. In the non-semantic (or direct) route, structural information activates the motor system directly, thus bypassing semantic memory. In this model, disruption of the semantic route would elicit a pattern similar to that exhibited by SD patients [39,40,41], whereas disruption of the non-semantic route would engender a pattern similar to that exhibited by apraxic patients [42].

Each time a user encounters a tool, he gains sensorimotor experience. With repeated interactions with the tool, these experiences will be accumulated in long-term memory where they facilitate the user’s activation of canonical motor action in future encounters with that specific tool. These sensorimotor representations are referred to as manipulation knowledge [35,48,49,50] or as the manipulation-based perspective. Assuming that the user is familiar with a tool, there will be corresponding manipulation knowledge available in long-term memory. However, as the saying goes, “All familiar things were once strange.” So how might an individual be able to use a tool if it is his first encounter with that tool and, therefore, no prior sensorimotor knowledge is available? 

For novel tools with no prior sensorimotor knowledge, structural information about the tool alone is of little use in activating the action production system. However, if the structural information is expressed exclusively in terms of the properties of the action system, visual input alone could elicit the intended actions, even in the absence of semantic knowledge. This possibility is exactly what Gibson [51,52] envisioned in his concept of affordances, which he introduced to account for the reciprocal interaction between the animal and the environment. Specifically, Gibson contended that an animal encountering the surrounding environment perceives a layout of surfaces scaled relative to the animal’s action capabilities rather than discrete object qualities (e.g., shape, size, texture, color, composition, mass, and motion) that are indifferent to the animal’s scale and action capabilities. Thus, affordances permit an animal to see the surroundings in terms of its action capabilities. For example, an object with a flat, rigid, extended, and knee-high surface affords sitting, whether the object is a chair in the dining room, a tree stump in the park, or even a swing hanging on a tree branch (specific object height, width, seat dimensions, armrest height, etc. are not required). However, a stool in the laundry room may afford sit-on-ability for an adult, but lean-body-on-ability for a toddler who has just started to stand. Thus, a specific object can have many different affordances, depending on the observer’s action capabilities and/or behavioral goals, for affordances are related, not only to the environment, but also to the observer. A chair, therefore, may also afford being used as a step stool (climb-on-able) or to hang a coat. 

To reiterate, for Gibson, the environment surrounding an animal abounds with opportunities for action, i.e., affordances. As scaled to the observer’s action capabilities, the perceived affordances contain the information necessary to calibrate motor parameters for successful execution of the intended action (see [48,53,54] for similar proposals). Unlike manipulation knowledge, which is stored as abstract representations in long-term memory, affordances are available directly as invariant patterns in the visual stimulation ambient at the point of observation. 

For proponents of the manipulation-based approach, tool use is processed along two channels. In cases of familiar tools, the visual features extracted from sensory information are further processed by manipulation knowledge stored in long-term memory, from which information about the movements associated with canonical manipulation is retrieved and then transmitted to the motor cortex for its execution. In cases of novel tools, on the other hand, the affordance properties described by Gibson (e.g., a hammer could present as affording characteristics of wieldable, graspable, pound-able, reachable, etc.), as opposed to its structural properties (e.g., size, weight, distance to the head of a nail, etc.) are submitted directly through the non-semantic route to the production system for appropriate motor outputs [46,49].

The reasoning-based perspective contends that tool use is an example of problem-solving in which the user accesses mechanical knowledge to reason about the physical properties of tools (e.g., a hammer with a flat and rigid head is sufficient to transfer power to a nail) and the target objects (e.g., a nail with a flat head can receive power when struck by the hammer) [32,55,56,57,58] The strength of this hypothesis is that the process it describes is applicable to any tool use case (i.e., the use of novel tools with no associated semantic memory, the prototypical use of familiar tools, and the unusual use of familiar tools), irrespective of users’ familiarity with tools. Often, however, the same tools are used repeatedly (e.g., alternating a spoon and chopsticks during a meal). Irrespective of repetitive tool use, the reasoning-based model initiates the same computational processes in every encounter with a tool, a highly implausible way of using tools. The manipulation-based account, by contrast, activates the same memory storage, thus avoiding recreating the same computational process with each encounter. Any advantage of the reasoning-based model over the manipulation-based model with respect to novel tool use becomes a disadvantage with respect to familiar tool use. 

To date, empirical evidence has tended to support the reasoning-based hypothesis [59,60,61]. Sensorimotor knowledge, or what Gibson [51,52] referred to as affordance perception, has, to date, been offered only as a post hoc description [39,40]. Consequently, we continue to lack (specifically in the manipulation-based camp) evidence demonstrating “how patients perceive affordances as usually assessed by the ecological approach to visual perception” [56]. 

### The Present Study

Praxis disturbance, the predominant symptom in patients with left brain damage (LBD) after stroke, is also prevalent in various neurodegenerative disorders including AD. In fact, apraxia is a diagnostic feature of AD and is included in the diagnostic guidelines of the National Institute of Neurological and Communicative Disorders and Stroke-Alzheimer’s Disease and Related Disorders Association (NINCDS-ADRDA) for probable or possible AD [62]. Yet praxis disturbance has failed to garner interest among researchers. 

As described above, there are two competing hypotheses as regards tool use behavior—the manipulation-based and reasoning-based approaches. In the reasoning-based account, it is impairment of mechanical knowledge that causes apraxia of tool use. Based on their literature review of studies directed at tool use disorders in LBD patients, Baumard et al. [59] concluded that failure in mechanical knowledge underpins tool use deficit in LBD patients. Lesourd et al. [61] examined whether impairment of tool use in AD and SD is of the same nature as that observed in LBD. When given mechanical problem-solving tasks, AD patients, despite experiencing difficulties, engaged in strategies such as trial-and-error to solve problems, a pattern not observed in LBD patients. Based on their findings, the authors concluded that impairment of mechanical knowledge is not what causes tool use deficit in AD. However, they left open the question of the source of tool use impairment in AD patients.

In the tool use research described above, proponents of the manipulation-based approach, in particular, conjecture that when using an unfamiliar tool, the user relies on affordances of the object as specified in the ambient optic array. Because affordances are scaled to the action capabilities of the user, they provide the information necessary for the control and coordination of movement. As invariant patterns contained in the structured arrangement in the ambient light at the observation point, affordances can be perceived by an animal with a suitable optical apparatus while occupying a particular observation point. Thus, despite whether a pair of wooden sticks on the table is familiar (used before at Chinese restaurants) or unfamiliar (not used before), we can easily pick one stick to stir hot chocolate mix in the cup or to roast a marshmallow. If our perceptual capacity is somehow disturbed, however, we will fail to perceive affordances, which could be the cause of the tool use deficit seen in AD. 

To date, the affordance hypothesis in tool use has been suggested solely as a post hoc explanation and has never been validated empirically. The conclusion by Lesourd et al. [61] ruling out disturbance in mechanical knowledge as the source of tool use deficit in AD can be construed as indirect evidence (by default) corroborating affordance perception impairment as the cause of tool use deficit in AD. Direct validation would be preferable, and that is the primary goal of this study. As a first step in assessing the feasibility of the affordance hypothesis in tool use, we examined whether the capacity to perceive affordances is impacted by AD. If this capacity is indeed disrupted by AD, this would further reinforce the preceding evidence that affordance perception capacity subserves tool use in AD. 

Another objective of the present study was to find additional measures that can enhance the sensitivity of current neuropsychological tests to detect subtle cognitive changes, even in the asymptomatic stage of AD. To assess whether defective affordance perception capacity provides the needed diagnostic power, we included patients with MCI in the study. MCI refers to a group of individuals who have some cognitive impairment, but the impairment is of insufficient severity to constitute dementia [63], thus representing a transitional stage between normal aging and dementia. However, with an annual conversion rate of 10–15%, patients with MCI are at high risk of progressing to AD [64]. Despite the significant risk of developing AD, there are currently no recommended diagnostic criteria to confirm MCI. At present, MCI is largely diagnosed by clinicians’ judgment based on the results of various neuropsychological tests complemented by laboratory tests and imaging data. If we can identify affordance perception deficits specific to AD, they might also be specific to MCI, thereby providing a potential cognitive marker for MCI. 

Parkinson’s disease (PD) is a neurodegenerative disorder caused by degeneration of dopaminergic neurons within the substantia nigra that results in dopamine depletion in striatum [65,66]. As the second most common neurodegenerative disorder after AD, PD is characterized by four primary symptoms (tremor, rigidity, bradykinesia, and postural instability) that affect motor control. Although not as prevalent as in AD, several studies have reported apraxic deficits in PD [67,68,69,70]. However, based on the high correlation between limb apraxia and visuospatial dysfunction, Burrell et al. [71] (also [67]) caution that the limb apraxia observed in PD may have occurred due to visuospatial dysfunction. If so, apraxic disturbances observed in AD and PD may have different causal underpinnings. Thus, defective affordance perception capacity, which we suspect to underlie praxic disturbance in AD, should be able to discriminate between these two most common neurodegenerative diseases. To assess this possibility, we also included patients with PD in the study.

In research on tool use, three different aspects of tool use have been highlighted—using novel tools with no associated semantic memory, using familiar tools, and applying familiar tools in a novel way. In the current study, we assessed praxis disturbance (i.e., defective affordance perception) as participants’ ability (or lack thereof) to identify an alternative use for a familiar tool. Today our surroundings are peppered with artifacts designed to carry out specific functions. When encountering well-designed artifacts, the designer’s intended affordances are easily recognized. However, human artifacts often provide more than one affordance because of their physical properties (e.g., shape, size, weight, and material composition) [72,73] For example, chopsticks, originally designed as eating utensils, can also function as skewers, stirrers, or even drumsticks, depending on the user’s needs.

It is possible to subsume a set of diverse objects, each with a different primary affordance, under the same secondary affordance. For example, a bowl, a jam jar, a shoe, or even a safety helmet can all support (afford) scooping water from a brook. For that reason, a secondary affordance can serve as an effective method to assess an individual’s capacity to detect an alternative use for a familiar tool. Kim and Kim [72] confirmed the efficacy of this procedure for assessing the ability of patients with schizophrenia to identify multiple affordances. 

Because participants in the current study were older adults and patients with dementia, we simplified our experimental procedures to facilitate their cooperation and task completion. In particular, we employed a single response, Go/No-Go paradigm to minimize the demand on working memory and verbal or numerical ability. In addition, we kept the experimental duration brief and comprised only of three short blocks to avoid participant fatigue.

AD is also known to impact visual sensory pathways by causing a variety of visual dysfunctions (see [74] for review). If AD patients perform poorly in the affordance perception task, other contributing factors must be ruled out, of which visual processing deficit is one. To rule out visual processing deficit, in Experiment 2, which used the same images of objects as those in Experiment 1, participants were asked to identify the objects’ physical properties (e.g., shape, color, or material composition) instead of their functions.

## 2. Experiment 1: Affordance Perception

### 2.1. Materials and Methods

#### 2.1.1. Participants

Twenty-two AD patients (8 males and 14 females), 22 MCI patients (6 males and 16 females), 21 PD patients (11 males and 10 females), and 17 healthy elderly control (EC) participants (5 males and 12 females), a total of 82 participants participated in the study. The data collected were analyzed using a mixed-design analysis of variance (ANOVA) with one between factor (4 participant groups) and one repeated measure (see Data Analysis section for more details). A G*Power(company, city and country.) analysis estimated a sample size of 68 for this design to reach at least a medium effect size of f = 0.3, 1-β = 0.8, and α = 0.05. AD, MCI and PD patients, all enrolled in Kyungpook National University Hospital’s outpatient clinic, volunteered for the experiment. All participants had normal or corrected-to-normal vision and reported no history of ophthalmologic disorder.

AD patients were selected on the basis of the diagnostic guidelines of the NINCDS-ADRDA for probable or possible AD [62]. The selection of MCI patients was based on the diagnostic guidelines of Petersen’s criteria for MCI [63]. Additional evaluations included neurological examinations, laboratory blood tests, and either an CT or MRI scan to exclude other causes of dementia. The diagnosis of PD patients conformed to the UK Brain Bank Criteria. PD patients’ overall motor symptoms, as measured by the Hoehn & Yahr stage scale, varied from 1.5 to 3 (*M* = 2.30, *SD* = 0.44) during on-period. Elderly controls (EC) were volunteers or the relatives of patients and were all in good mental and physical health. 

Dementia severity was assessed by the Korean adaptation [75] of the Mini Mental State Examination (K-MMSE) [76]. The result revealed significant differences among the four participant groups, *F*(3, 78) = 17.39, *p* < 0.001. A Tukey post hoc test revealed that the AD group differed from the other three participant groups at the 0.05 level. Clinical Dementia Rating (CDR) [77] scores were also collected from AD patients. Their scores varied from 0.5 to 1 (mean CDR = 0.98, *SD* = 0.10). The four participant groups were matched for years of education, *F*(3, 78) = 2.01, *p* > 0.05, but not for age, *F*(3, 78) = 4.55, *p* < 0.01. A Tukey post hoc test revealed two homogeneous subsets with AD and MCI in one subset and MCI, PD, and EC in the other. Demographic data for the participants are presented in Table 1.

#### 2.1.2. Apparatus

Color images of 18 household items served as the stimuli for the present experiment (Figure 1). All images were sized to 600 × 800 pixels and presented on a 15-inch laptop with a pixel resolution of 1024 H × 760 V. The presentation of stimuli was controlled by DirectRT (Empirisoft Corporation, 2012), which also recorded responses and measured accuracy and reaction times of the responses. Participants viewed the display binocularly at a distance of approximately 50 cm.

#### 2.1.3. Stimuli

Each artifact exhibited its designed affordance clearly. The procedure for selecting the artifacts used in the present study was the same as that adopted by [72] which, in turn, was a slight modification of that adopted by [73]. Three pairs of affordances were used: (a) scoop-with/pierce-with; (b) pour-in-able/stretchable; (c) cut-able-with/mop-up-with. 

As shown in Figure 1, the six objects selected to evaluate each affordance pair were divided into two mutually exclusive classes: O_aff 1_ had the first affordance (e.g., scoop-with) but not the second (e.g., pierce-with); O_aff 2_ had only the second affordance, but not the first. 

#### 2.1.4. Design

The experiment consisted of three randomized blocks of 72 trials. Each block consisted of 6 images of objects (3 of O_aff 1_ and 3 of O_aff 2_). Each object was presented twice for a total of 12 trials. In each pair of affordances, one served as the target signal and the other as the distractor. After 12 trials, the same procedure was repeated, but with the Go/No-Go categories reversed. This manipulation yielded a 2 (Affordance Pair: O_aff 1_ & O_aff 2_) × 3 (Object) × 2 (Repetition) × 2 (Signal/Distractor) for a total of 24 trials for each affordance pair.

#### 2.1.5. Procedure

Prior to the experiment, all participants completed the informed consent form and then completed the MMSE test. Participants were tested individually. 

Participants were told that household items can be used to perform various functions other than their prototypical (designed) function. For example, the experimenter demonstrated that a paper cup designed to contain liquid could also serve as a candle holder. Using this example, participants were told to press the space bar as quickly as possible if the displayed object had the specific affordance (function) and was therefore a “go” stimulus) but withhold a response (i.e., not pressing the space bar) if an object did not represent the target affordance (function) and therefore constituted a “no-go” stimulus. No-Go trials were terminated after a 5 s timeout. 

After 12 trials, the same procedure was repeated by reversing the signal/distractor categories. Prior to the initiation of each half-block, a text message appeared on the computer screen informing the participant of the target function of the objects they would view, and the experimenter explained the target functions verbally using the basic descriptions of [73], but with a slight modification to better fit the Korean culture (see [72] for further details).

A practice session was created using representative objects for each affordance pair: a gourd dipper for scoop-with-able vs. a nail for pierce-with-able; a mug for pour-in-able vs. rubber bands for stretchable; and a plastic knife for cut-with-able vs. a towel for mop-with-able. The two objects in each affordance pair appeared twice in a practice set. Each practice set was repeated until the participant demonstrated that he/she fully understood the procedure. A similar 4-trial practice session preceded each block of the experiment.

### 2.2. Data Analysis

Performance of the four groups was compared for accuracy and reaction time. Each block was comprised of 24 trials produced by a 2 (Affordance Pair: O_aff 1_ & O_aff 2_) × 3 (Object) × 2 (Repetition) × 2 (Signal/Distractor) design. With a pair of affordances constituting each block, one (i.e., O_aff 1_) was the target whereas the other (O_aff 2_) served as a distractor. Thus, a hit or a correct rejection was coded as a correct response and a miss, or a false alarm was coded as an incorrect response. 

While inspecting the data, we noticed that several participants were 50% accurate with all reaction times fixed at 5 s timeout. Upon inspection of their data, we realized that these participants withheld their responses throughout the trials. Despite our efforts to familiarize participants with the task and the procedure in the practice session, these participants were not ready to proceed to the experiment. It is worth noting that the elderly population of Korea has had limited experience with (laptop) computers. Some participants were unable to get accustomed to pressing the enter key to trigger the display and then responding by pressing the space bar. Ultimately, we excluded 7 participants’ data (6 AD patients, and 1 MCI patient) from analysis who withheld their responses for more than 3 half sessions (from a total of 6 half or 3 full sessions). In the revised demographic data, the four groups were matched for age, *F*(3, 71) = 2.64. *p* > 0.05, and years of education, *F*(3, 71) = 1.50, *p* > 0.05, but not for MMSE, *F*(3, 71) = 11.97, *p* < 0.001. A Tukey post hoc test differentiated the AD group from the other three groups at the 0.05 level.

### 2.3. Results and Discussion

For the remaining participants whose data were entered into analysis, there seemed to be a tendency toward withholding responses. To explore this possibility, the entire 72 trials were divided into two response categories (target vs. distractor) by combining across all six half-blocks after collapsing across 3 objects. Given that we employed a simple design with a relatively small number of trials, pulling responses across the independent variables provided the additional benefit of increasing statistical power. 

The results were entered into a 2 (Go/No-Go) × 4 (Group) mixed design ANOVA. The ANOVA demonstrated a significant main effect of Go/No-Go, *F*(1, 71) = 126.47, *p* < 0.001, η_p_^2^ = 0.64. Accuracy for the Go trials was 73% and for the No-Go trials was 98%. The ANOVA further confirmed a main effect of Group, *F*(3, 71) = 5.97, *p* < 0.01, η_p_^2^ = 0.20, and a significant interaction between Go/No-Go and Group, *F*(3, 71) = 4.17, *p* < 0.01, η_p_^2^ = 0.15 (Figure 2). A simple effects analysis demonstrated that the effect of Group on the Go trials was significant, *F*(3, 71) = 5.31, *p* < 0.01, but not on the No-Go trials, *F*(3, 71) < 1, *ns*. A Tukey post hoc test on the group effect using performance on the Go trials revealed two homogenous subsets with AD and MCI in one subset and MCI, PD, and EC in the other. 

A one-way ANOVA on the Group effect using reaction time on the Go trials was also reliable, *F*(3, 71) = 4.16, *p* < 0.01 (Figure 3). A Tukey post hoc test replicated the same pattern observed in the accuracy data, i.e., two homogeneous subsets with AD and MCI in one subset and MCI, PD, and EC in the other. 

To summarize briefly, the performance of the four groups on the Go trials was assessed using accuracy and reaction time. The results demonstrated that AD patients (59%, 1.75 s) performed poorest, followed by MCI (74%, 1.55 s), PD (78%, 1.27 s) and EC (82%, 1.24 s), although the difference between the PD and EC groups did not reach statistical significance. Further analysis revealed that AD patients performed only at chance level (50%), *t*(15) = 1.76, *p* > 0.05, suggesting that they responded randomly to the stimuli. 

## 3. Experiment 2: Physical Property Detection

There is a possibility that a visual processing deficit, prevalent in AD, might have contributed to the poor performance observed in the AD patients and the slightly degraded performance of MCI patients. Experiment 2 was designed to rule out this possibility. Participants were asked to identify objects’ physical properties (e.g., shape, color, or material composition) instead of their functions. The same images of objects and the same procedure used in the previous experiment were used in this experiment. To be specific, participants judged whether the displayed object contained a certain color (e.g., green), a certain shape feature (e.g., a rounded shape), or a certain material (e.g., fabric). Six physical properties—two colors (pink and green), two shapes (right angle and rounded shape), and two types of material (fabric and wood)—were chosen for this purpose. As in the previous experiment, the experiment was composed of 3 randomized blocks, with each block comprising of a pair of mutually exclusive physical properties in which one served as the target and the other served as a distractor. 

### 3.1. Participants

The same 82 participants who participated in Experiment 1 participated in Experiment 2. Data from the 7 participants who withheld responses in Experiment 1 were excluded from analysis.

### 3.2. Apparatus and Stimuli

The same apparatus and stimuli used in the previous experiment were used.

### 3.3. Design

The experiment consisted of three randomized blocks with each block comprising a pair of physical properties. The paired properties were: (a) pink/right angle, (b) fabric/circular shape, (c) green/wooden material. As in Experiment 1, each block was comprised of two half blocks. In each half-block, one of the physical property pairs was chosen randomly as the target and the other as a distractor. Their roles were reversed in the second half-block. The objects chosen for each physical property are listed in Table 2. The six objects were presented twice, resulting in a total of 12 trials, which were repeated after reversing the target and distractor. Thus, the 24 randomized trials constituting each block were produced by a 2 (Physical Property: P_1_, P_2_) × 3 (Object) × 2 (Repetition) × 2 (Target/Distractor) design.

### 3.4. Procedure

The same procedure used in Experiment 1 was used for Experiment 2. As in Experiment 1, a 2-trial practice session preceded each block of the experiment. The representative images that were used to produce practice trials in each block were pink roses (pink), a window (right angle), cloth (fabric), a coin (circle), a green bell pepper (green), and a wooden toy dog (wood). Prior to the initiation of each half-block, instructions were displayed in text on the computer screen informing participants about the target property in the displayed objects.

### 3.5. Data Analysis

As in Experiment 1, to examine whether participants had a similar tendency to withhold responses in this experiment as in the previous one, responses were divided into Go and No-Go trials by combining responses from all six blocks after collapsing across property and entered into 2 (Go/No-Go) × 4 (Group) mixed design ANOVA. In addition, responses on the Go trials were compared with those from Experiment 1 to assess whether our findings from Experiment 1 suggest defective affordance perception capacity or visual processing deficit. 

### 3.6. Results and Discussion

The ANOVA confirmed a main effect of Go/No-Go, *F*(1, 71) = 38.50, *p* < 0.01, η_p_^2^ = 0.35. The Go trials were 86% accurate, whereas the No-Go trials were 95% accurate. The Group effect was again statistically significant, *F*(1, 71) = 4.87, *p* < 0.01, η_p_^2^ = 0.17. A Tukey post hoc test revealed two homogeneous subsets with AD and MCI in one subset and MCI, PD, and EC in the other. 

A one-way ANOVA on reaction time data from the Go trials also revealed a significant effect of Group, *F*(3, 71) = 3.85, *p* < 0.05 (Figure 3). A Tukey post hoc test again divided the patient groups into two with MCI, PD, and EC in one subset and MCI and AD in the other. 

The main effect of Go/No-Go suggests that participants still tended to withhold responses on the Go trials, but this tendency was weaker than that observed in Experiment 1. To better evaluate the potential differences among groups, an ANOVA with Group and Experiment (Experiment 1 vs. 2) as the independent variables and responses on the Go trials from the two experiments as the dependent variable was conducted. The effect of Experiment was reliable, *F*(1, 71) = 44.64, *p* < 0.001, η_p_^2^ = 0.39 (Figure 4). The main effect of Group was also significant, *F*(3, 71) = 6.25, *p* < 0.01, η_p_^2^ = 0.21. A simple effects analysis on Group confirmed significant differences in tendency between the two experiments for AD patients, *F*(1, 71) = 25.36, *p* < 0.001, η_p_^2^ = 26, for MCI patients, *F*(1, 71) =8.67, *p* < 0.01, η_p_^2^ = 0.11, and for PD patients, *F*(1, 71) = 12.11, *p* < 0.01. η_p_^2^ = 15. However, this difference did not reach significance for EC, *F*(1, 71) = 12.33, *p* > 0.05. The improved performance by AD patients was apparent. In fact, whereas AD patients responded in a random manner to the Go trials in Experiment 1, their responses in the same condition in the current experiment was much more systematic, achieving 80% accuracy, well above chance level, *t*(15) = 9.51, *p* < 0.001. 

An ANOVA with Experiment and Group as independent variables demonstrated a main effect of Experiment, *F*(1, 71) = 79.63, *p* < 0.001, η_p_^2^ = 53. Participants from all 4 groups responded quicker in Experiment 2 (*M* = 1.2 s) than in Experiment 1 (*M* = 1.45 s). In fact, reduction in RT was reliable in each of all 4 groups [*F*(1, 71) = 18.03, *p* < 0.001, η_p_^2^ = 0.20 for AD; *F*(1, 71) = 20.44, *p* < 0.001, η_p_^2^ = 0.22 for MCI; *F*(1, 71) = 26.31, *p* < 0.001, η_p_^2^ = 0.27 for PD; *F*(1, 71) = 15.58, *p* < 0.001, η_p_^2^ = 0.18 for EC]. Yet, the main effect of Group was still significant, *F*(3, 71) = 4.27, *p* < 0.01, η_p_^2^ = 0.15. The four groups formed two subsets, with MCI, PD, and EC in one subset and MCI and AD in the other subset, replicating the same division observed in the previous post hoc tests. 

To summarize, performance improved drastically across all 4 groups in Experiment 2. Particularly notable were AD patients who responded almost in a random fashion in Experiment 1 but responded in a systematic fashion, achieving performance well above chance level in Experiment 2. Given these results, we can conclude that visual processing deficit is unlikely to have caused the poor performance of AD patients and the slightly degraded performance of MCI patients in Experiment 1.

## 4. General Discussion

Praxis disturbance, although prevalent in AD, has been largely neglected by AD researchers. Particularly disturbed in AD patients is their ability to use tools, also called apraxia of tool use. Given the urgent need for non-invasive, cost effective, and easily available diagnostic tools for early detection of AD—even during otherwise asymptomatic stages, we set out to explore whether evaluating patients for tool use disturbance might boost the diagnostic power of existing neuropsychological tests so that the tests can detect more subtle alterations in cognition, even in the earliest stages of AD. 

To date, two dominant and competing hypotheses attempt to account for variations in tool use behavior. The manipulation-based account relies on semantic memory, which is accumulated through prior sensorimotor experiences with a particular tool. In this view, an individual, when using a familiar tool, retrieves information from stored sensorimotor experiences (manipulation knowledge) about the tool’s purpose, its target object, and the typical movement associated with the tool. The reasoning-based account views the current situation a potential tool user faces as an instance of problem-solving in which she uses mechanical knowledge to reason about the structural properties of tools and their action recipients to solve the current problem. 

Apraxia is most common in patients with left hemisphere damage after a stroke. It is well documented that mechanical knowledge is severely disturbed in LBD patients [59]. Lesourd et al. [61] investigated whether the same loss of mechanical knowledge causes impairment of tool use in AD. Inspection of strategy profiles generated based on each participant’s handling of the problem revealed the absence of strategies in LBD patients, but the use of strategies similar to controls in AD patients. Based on these findings, the authors rejected mechanical knowledge impairment as the source of tool use deficit in AD but left the answer open. 

Proponents of the manipulation-based account of tool use suggest that, in cases involving novel tools, it may be affordances, a concept proposed by Gibson [51,52]. that facilitate their applications. For Gibson, our understanding of the surrounding environment is expressed in reference to our action capabilities, i.e., as affordances (opportunities for action). The environment abounds with opportunities for action. Significantly, these environmental properties are readily available as invariants in the energy distributions ambient around an observation point. An affordance, when perceived, directly maps onto the motor parameters of the action production system and thus is immediately ready to be implemented into an intended action. 

To date, the idea that novel tool use may be mediated through affordances has been offered only as a post hoc description and has not been scientifically validated [39,40]. The present study took the first step towards validating the utility of affordance as a simpler explanation of tool use behavior. Tool use behavior occurs in three distinct formats—use of novel tools, use of familiar tools, and use of familiar tools in novel ways. Because we are surrounded by man-made artifacts, each of which carry multiple (secondary) affordances other than the primary affordance (i.e., its designed function), we assessed participants’ capacity to perceive secondary affordances of various objects as a way to measure their capacity to identify alternative uses for familiar tools. 

An experiment employing a single response Go/No-Go paradigm was administered to four groups, AD, MCI, PD, and EC, with the groups matched for age and years of education. The AD group performed poorest, followed by MCI, and PD and EC, in that order. EC and PD groups performed comparably, their differences failing to reach statistical significance. The AD group responded randomly to stimuli, their performance not differing from chance. In Experiment 2, wherein participants judged physical properties, rather than functions, of objects, even AD patients performed reliably, well over chance level, thus ruling out deficit in visual processing as the basis for their poor performance in Experiment 1. Also notable was the performance by MCI patients. Although their level of performance did not differ significantly from those of the PD and EC groups, it also did not differ significantly from that of AD group, suggesting only slight degradation in MCI patients’ performance. 

Finding a decline in perceptual capacity for affordances only in AD patients, but not in PD or EC groups, suggests that perceptual capacity is capable of differentiating AD from normal aging, as well as from other neurodegenerative disorders such as PD. Of interest to the present finding is a recent study reported in the literature demonstrating that the presence of apraxia and associated left parietal features are sufficient to differentiate AD from frontotemporal dementia (FTD) spectrum disorders [78,79]. Because of overlapping symptoms with AD, FTD spectrum disorders pose a significant challenge to existing diagnostic tools. Recall that Lesourd et al. [61] concluded that the tool use disorders observed in AD patients are of a different nature than those observed in LBD patients. In light of Lesourd et al.’s conclusion, we conjectured that the apraxic behavior referred to in the literature [78,79] and the tool use deficits investigated in the present study might share the same causal basis, that is, defective affordance perception capacity. In this regard, the present findings further reinforce those reported in [78,79]. Also encouraging is the finding that this perceptual capacity declines in the MCI stage. Although the exact causes of the pathophysiological process in AD are not yet clear, changes in affordance perception, albeit subtle, are detectable in the MCI stage, suggesting its potential role as an early indicator of AD—even in the preclinical stages.

It would be imprudent to overgeneralize the present findings. Of the potential issues that can be raised with this study, the most obvious is the small sample sizes represented by the four participant groups. Further, the age and lack of prior computer experience of study participants dictated our use of a simplified experimental design. Second, affordance perception capacity was evaluated using six types of affordances. Clearly, this list needs to be expanded to increase the generality of the affordance hypothesis for tool use behavior. Third, participants’ age and diagnoses dictated our keeping the experiment brief, so we did not conduct a praxis assessment of participants. Such data, particularly, from patients with AD and MCI could have served as the basis for evaluating the performance of the two patient groups in the present study.

## 5. Conclusions

Taken together, the present findings support a strong argument that disturbance of affordance perception may serve as an additional defining feature of AD. As such, disturbance of affordance perception holds considerable promise, not only to enhance the diagnostic precision of neuropsychological testing, but also to provide an inexpensive, non-invasive, and affordable tool for detecting AD, even in the disease’s earliest stages.

## Figures and Tables

**Figure 1 healthcare-10-00839-f001:**
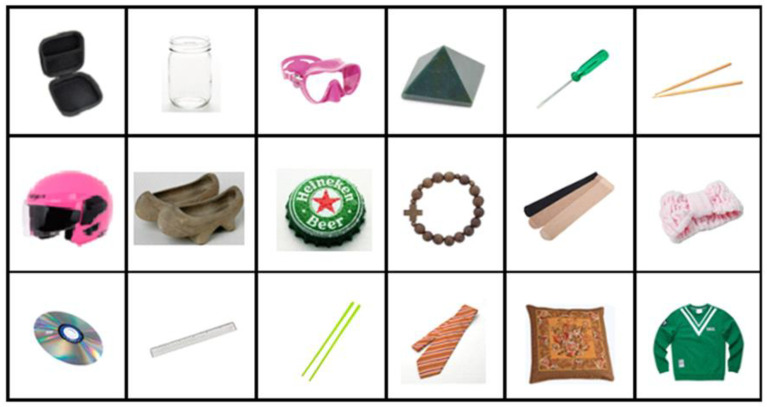
Images of objects used in Experiment 1. Top Row-Left 3: Objects with scoop-with affordance (earphone case, glass bottle, divers’ goggles), Top Row-Right 3: Objects with pierce-with affordance (polyhedron, screwdriver, drumsticks); Middle Row-Left 3: Objects with pour-in-able affordance (bicycle helmet, clogs, bottle cap). Middle Row-Right 3: Objects with stretchable affordance (rosary, stockings, cloth headband) Bottom Row-Left 3: Objects with cut-able-with affordance (CD, plastic ruler, chopsticks)—Bottom Row Right 3 Objects with mop-up-with affordance (necktie, cushion, shirt).

**Figure 2 healthcare-10-00839-f002:**
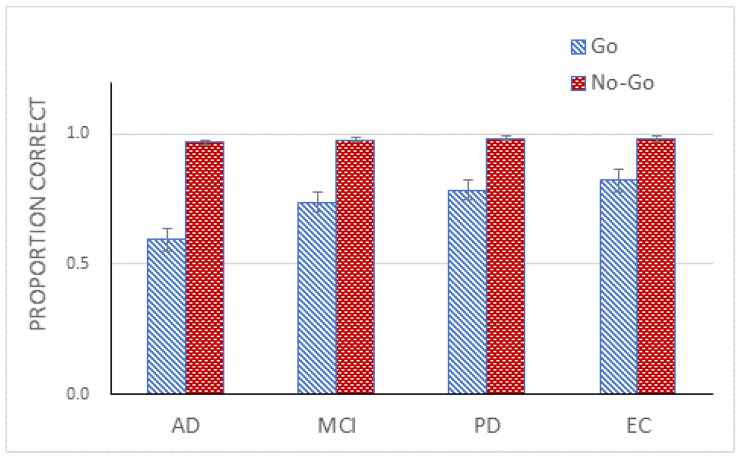
Mean proportion correct (with standard error bars) for four participant groups for the Go and No-Go trials in Experiment 1.

**Figure 3 healthcare-10-00839-f003:**
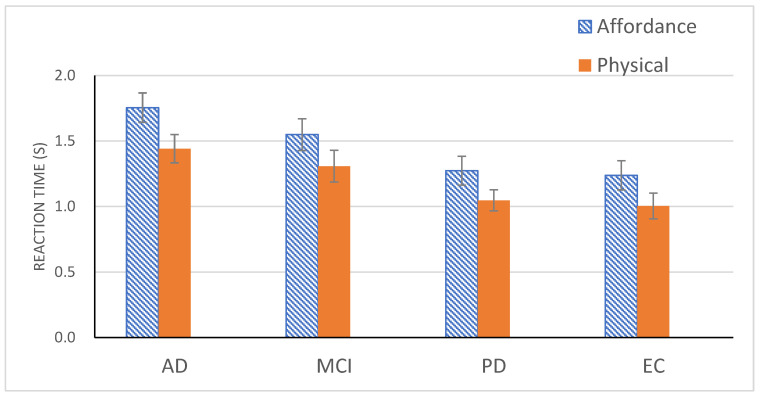
Mean reaction time (with standard error bars) for four participant groups for the Go trials in Experiment 1 (affordance detection) and for the Go trials in Experiment 2 (physical property detection).

**Figure 4 healthcare-10-00839-f004:**
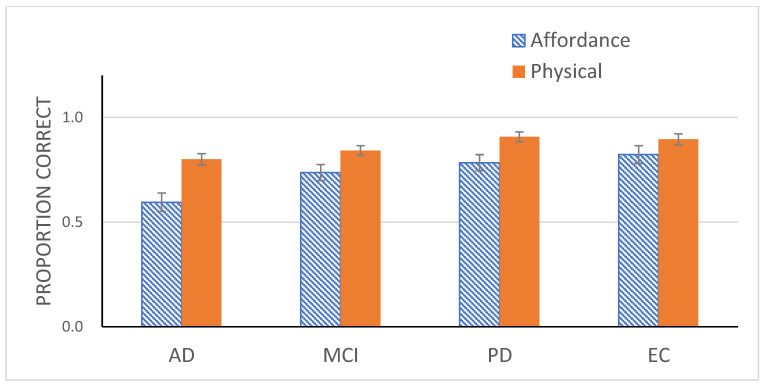
Mean proportion correct (with standard error bars) for four participant groups for the Go trials in Experiment 1 (affordance perception) and for the Go trials in Experiment 2 (physical property detection).

**Table 1 healthcare-10-00839-t001:** Demographic data of participant groups.

	EC (n = 17)	AD (n = 22)	MCI (n = 22)	PD (n = 21)
Age (years) *	67.4 ± 9.3	74.0 ± 6.6	70.1 ± 6.8	66.1 ± 7.2
Edu (years)	9.5 ± 3.6	8.2 ± 5.1	7.4 ± 5.4	10.4 ± 2.9
MMSE ^+^	27.2 ± 2.5	20.1 ± 4.7	24.4 ± 3.4	26.8 ± 2.9

Notes: Data presented as mean ± SD. * significant at *p* < 0.05; ^+^ significant at *p* < 0.01. Abbreviations: EC, elderly controls; AD, Alzheimer’s disease; MCI, mild cognitive impairment; PD, Parkinson’s disease; MMSE, Mini Mental State Examination.

**Table 2 healthcare-10-00839-t002:** Physical properties of objects used in Experiment 2.

Physical Properties	Objects		
Pink	bicycle helmet	cloth headband	diver’s goggles
Right angle	plastic ruler	cushion	polyhedron
Fabric	stocking	necktie	earphone case
Circle	bottle cap	compact disk (CD)	glass bottle
Green	shirt	screwdriver	chopsticks
Wood	clogs	drumstick	rosary

## Data Availability

The data presented in this study are available on request from the corresponding author.

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
