# Peer review of "Impaired Affordance Perception as the Basis of Tool Use Deficiency in Alzheimer’s Disease"

_healthcare, 2022, doi:10.3390/healthcare10050839_

Round 1

Reviewer 1 Report

Congratulations for the scientific effort made by your team to addressing this interesting question.

  1. What is the power of this research study, how they calculated sample size: elaborate in 1-2 sentence?
  1. On what basis do no. subjects are taken in any given group, is there any reference article please explain?
  1. In a volunteer group which database have been used for their mental and physical health?
  1. What are the limitations of the study, explain in one paragraph?
  1. How this study will be helpful in future research because authors have taken three different disease group comparing with same control?
  2.  
  3.  Add the crisp paragraph of your study conclusion

Author Response

Congratulations for the scientific effort made by your team to addressing this interesting question.

Responses: Thank you for your compliment.

What is the power of this research study, how they calculated sample size: elaborate in 1-2 sentence?

Responses:  Per request, we added the following sentences as a footnote in the participants subsection:

“Note that the data collected were analyzed employing a mixed-design analysis of variance (ANOVA) with one between factor (4 participant groups) and one repeated measure (see Data Analysis section for more detail). A G*Power analysis estimated a sample size of 68 for this design to reach at least a medium effect size of f = 0.3, 1 - b = 0.8, and a = 0.05” in the participants subsection.

On what basis do no. subjects are taken in any given group, is there any reference article please explain?

Responses:  In selecting participants for our study, we followed the guidelines described in the participants’ subsection, i.e., the NINCDS-ADRDA for AD and Petersen’s (2004) for MCI. Additional lab tests augmented by MRI or CT scans were administered in diagnosing the patients.

In a volunteer group which database have been used for their mental and physical health?

Responses:  Patients who volunteered for our study (AD, MCI, PD) were all outpatients of the university hospital with which the third author is affiliated. Please note that there are no healthcare databases maintained in Korea.

Please note that we administered MMSE to screen participants with cognitive impairment. In addition, given the nature of the task performed, we accepted only those participants with normal or corrected-to-normal vision

What are the limitations of the study, explain in one paragraph?

Responses:  We pointed out in the next to the last paragraph at least three potential issues that caution any attempts to overgeneralize the findings we report in this study.

How this study will be helpful in future research because authors have taken three different disease group comparing with same control?

Responses:  This study should be considered as a preliminary study. In the future studies we intend to address the limitations we identified in the conclusion section.

 Add the crisp paragraph of your study conclusion

Responses:  In the final paragraph, we stated clearly that the present findings are expected to contribute to improve diagnostic precision of neuropsychological testing, even in the early stages of AD.

Reviewer 2 Report

The introduction seems to suit more for a review article, distancing from the real focus of the work. In my opinion need to be completely reorganized, to be more clear and focused.

About the AD literature, I feel that works from A. Bush group, and from N. A. Rey group, which are developing new strategy to prevent cognitive impairment must be added in the present work introduction.

General Discussion must be improved to sound more clear and with a strong conclusion.

lines 40-46, 67-69, 79-86, 114-123 there are no reference for the reported informations

lines 92-95, 99-106 are too confused, please rewrite the sentences.

Figure 1 could be improved, is presented in the same scale that was presented for the paciennts? this must be better explained in the text

Figure 2 the graph report No-go, must be correct to No-Go, as reported in the text.

General corrections:

number% must be corrected as number %

Author Response

The introduction seems to suit more for a review article, distancing from the real focus of the work. In my opinion need to be completely reorganized, to be more clear and focused.

Responses:  In hindsight, we were too overjealous attempting to provide a comprehensive picture of the current state of Alzheimer’s disease. In the revision, we shortened the introduction substantially. Specifically, in the original submission the introduction consisted of 370 lines. In the revision, it is cut to 310, a reduction of 60 lines.

About the AD literature, I feel that works from A. Bush group, and from N. A. Rey group, which are developing new strategy to prevent cognitive impairment must be added in the present work introduction.

Responses:  In the revision, as we pointed out above, we shortened the introduction substantially. Included in that deletion was the discussion on AD drug development. While Prof. Ashley Bush’s research on AD drug development is tantalizing, we decided not to resurrect this issue and expand the intro again.

General Discussion must be improved to sound more clear and with a strong conclusion.

Responses:  In the final paragraph, we have listed the limitations of our study as well as its contribution.

lines 40-46, 67-69, 79-86, 114-123 there are no reference for the reported informations

Responses:  Per request, we added the following references:

lines 40-46: Mattap et al., 2022

lines 67-69: Dubois et al., 2007, 2016; Sperling et al., 2011

lines 79-86: Crous-Bou et al., 2017; de Bruijn et al., 2015

lines 114-123: Monsell et al., 2014; Bastin & Salmon, 2014; Belleville et al., 2014;

Short: Casaletto & Heaton, 2017; Salmon, 2019; Weissberger et al., 2017

lines 92-95, 99-106 are too confused, please rewrite the sentences.

Responses: line 92-95: this sentence is revised as “these biomarkers can only be obtained using lumbar puncture and PET scans. Both procedures are expensive, invasive and not widely accessible”

lines 99-106: these lines are deleted in the revision.

Figure 1 could be improved, is presented in the same scale that was presented for the paciennts? this must be better explained in the text

Responses: I assume you mean “patients.” Presenting the images in the same scale that participants saw them would take up an entire page (or more). 

As for the stimulus images, as we described in the apparatus subsection in the methods section of Experiment 1, “All images were sized to 600 ´ 800 pixels and presented on a 15-inch laptop with a pixel resolution of 1024 H ´ 760 V.”

Figure 2 the graph report No-go, must be correct to No-Go, as reported in the text.

Responses: corrected. Thanks for pointing it out.

General corrections:

number% must be corrected as number %

Responses:  All the instances of % have been corrected as advised.

Reviewer 3 Report

Comment 1:

The Introduction is too long and need to be summarized; the main points should be:

1-The AD etiopathology and neural functions affected

2-The studies and investigation results already published related to the degraded affordance detection method 

3-The therapeutic role and possible implications on the early stages of AD progression 

There is no need to add headings in this section.

Comment 2: 

 For clarity and transparency, Figures must include a scatter plot to depict the distribution of individual values obtained from each individual plus the mean and SE.

Comment 3: 

Figures should be renumbered since there are two Figure 1 in the Experiment 1 section. 

Comment 4: 

The authors should ensure the keywords are previously mentioned in the abstract (e.g. Apraxia...)

Author Response

Comment 1:

The Introduction is too long and need to be summarized; the main points should be:

1-The AD etiopathology and neural functions affected

Responses:  AD and apraxia are notorious for their heterogeneity. Thus there is no consensus as to the neural substrates of these disorders. Moreover, it has been suggested that the present manuscript, as it stands, is too comprehensive and we have been advised to narrow its focus. For those reasons, we left this issue unaddressed as it is beyond the scope of the present study.

2-The studies and investigation results already published related to the degraded affordance detection method

Responses:  As we underscored throughout the manuscript, the affordance hypothesis was offered in the literature as a post hoc description, but has never been validated empirically. We believe that this study is the first attempt to validate the affordance perception deficit hypothesis as the possible cause for apraxic deficits in Alzheimer’s patients.

3-The therapeutic role and possible implications on the early stages of AD progression

Responses:  Because the present study was directed at improving the precision of existing diagnostic tools for AD, particularly in its early stages, not developing therapeutic treatments to cure the disease, we have deleted most of the discussions on drug development as advised, a point also raised by reviewer 1.

There is no need to add headings in this section.

Responses:  Headings have been deleted as advised.

Comment 2:

 For clarity and transparency, Figures must include a scatter plot to depict the distribution of individual values obtained from each individual plus the mean and SE.

Responses:  A scatter plot is a graphic tool used to visualize the relationship between two continuous variables. Our study compares the performance of four participant groups (AD, MCI, PD, and EC). Thus. the independent variable in our study was discontinuous, which is not suitable for portraying as a scatterplot. Please note that figures 2-4 include group means and SEs.

Comment 3:

Figures should be renumbered since there are two Figure 1 in the Experiment 1 section.

Responses: We refer to Figure 1 twice, once in line 431 and the second time in line 441.

Comment 4:

The authors should ensure the keywords are previously mentioned in the abstract (e.g. Apraxia...)

Responses:  All keywords are mentioned at least once in the Abstract.

Round 2

Reviewer 2 Report

Congratulations on the scientific effort made by your team to address all the reviewers' questions.

Author Response

Thank you for your review.